# An Efficient and Streamlined System for In Vitro Regeneration and Genetic Transformation of Paper Mulberry (*Broussonetia papyrifera*)

**DOI:** 10.3390/life16010078

**Published:** 2026-01-04

**Authors:** Fangyu Ye, Tong Ke, Shuiqing Deng, Lan Pan, Ming Tang, Wentao Hu

**Affiliations:** State Key Laboratory of Conservation and Utilization of Subtropical Agro-Bioresources, Guangdong Laboratory for Lingnan Modern Agriculture, Guangdong Key Laboratory for Innovative Development and Utilization of Forest Plant Germplasm, College of Forestry and Landscape Architecture, South China Agricultural University, Guangzhou 510642, China; yeeverleaf@163.com (F.Y.); ketong555@outlook.com (T.K.); dsq17875306802@163.com (S.D.); panlan@scau.edu.cn (L.P.)

**Keywords:** hybrid *Broussonetia papyrifera*, tissue culture, genetic transformation, regeneration system

## Abstract

In the present study, we developed an efficient and reproducible protocol for in vitro regeneration and *Agrobacterium tumefaciens*-mediated genetic transformation of *Broussonetia papyrifera* (L.) L’Hér. ex Vent. (paper mulberry) using leaf explants from a hybrid genotype. First, we optimized surface sterilization of leaf explants. Treatment with 0.6% (*w*/*v*) sodium hypochlorite for 8 min, followed by three rinses with sterile water and blotting on sterile filter paper, yielded a 33.60% explant survival rate and reduced contamination to 35.84%. Second, we refined the co-cultivation step for transformation using *A. tumefaciens* strain EHA105 carrying pCAMBIA1300-35S-eGFP. Leaf discs were infected for 20 min and co-cultured for 2 days on co-cultivation medium overlaid with sterile filter paper, which limited the overgrowth of *A. tumefaciens*. After co-cultivation, explants were transferred sequentially to callus induction, shoot induction, shoot multiplication, and rooting media supplemented with 250 mg·L^−1^ cefotaxime and 200 mg·L^−1^ Timentin, as well as 5.0 mg·L^−1^ hygromycin at a concentration that completely suppressed regeneration of non-transformed explants. Meanwhile, after transfer to the callus induction medium, eGFP fluorescence was detected in resistant calli as an initial screening for transformants. The integration and expression of the transgene were further confirmed by PCR and quantitative reverse transcription PCR (qRT-PCR) after the resistant calli developed into plantlets. Collectively, this streamlined protocol provides a practical platform for functional genomics and genetic improvement of *B. papyrifera*.

## 1. Introduction

*Broussonetia papyrifera* (L.) L’Hér. ex Vent. (paper mulberry) is a perennial deciduous tree in the family Moraceae. Its fruits, leaves, bark, and wood are used for nutritional, medicinal, and industrial purposes [1,2]. The fibre chemistry and structure are comparable to those of ramie (*Boehmeria nivea*) and cotton (*Gossypium hirsutu*m), which makes *B. papyrifera* a promising raw material for papermaking [3]. In addition to its value as a fibre source, *B*. *papyrifera* is a pioneer tree with a strong tolerance to adverse environments [4,5]. As a result, it is widely used for ecological restoration and the rehabilitation of degraded ecosystems [6,7]. Moreover, its rapid growth and high capacity for shoot regeneration also support fast establishment in plantations [8].

High-quality *B. papyrifera* seedlings are essential for applications in ecological restoration, forage production, and papermaking [9]. Several propagation methods exist for this species, including seed, stem cuttings, layering, root suckers, and in vitro tissue culture. Although seed germination is relatively slow, it can be improved by H_2_SO_4_ pretreatment [10]. In contrast, stem cuttings are widely used in research and practice; however, cutting-derived plants often exhibit uneven, poorly developed root systems [11]. Furthermore, propagation via root suckers can generate plants with well-developed roots and good field performance, but it is difficult to scale for large-scale seedling production [12,13]. Among these approaches, tissue culture stands out as it enables rapid clonal multiplication while preserving elite traits and genetic stability, which is advantageous for intensive nursery production [14]. Notably, regeneration systems have already been established for many model woody species, including *Eucalyptus*, poplar, and *Paulownia* [15,16,17]. Thus, optimizing propagation methods, especially tissue culture, remains a key step toward broader application of *B*. *papyrifera* across diverse fields.

Regeneration protocols have been reported for Hybrid *B. papyrifera* and for *B*. *kazinoki* introduced from Japan, and most methods rely on leaf explants [18,19]. However, the rough leaf surface of *B. papyrifera* readily harbours microorganisms, which increases the risk of contamination during culture. For example, Wang (2010) reported a sterilization procedure using 70% ethanol for 20 s, followed by 0.1% HgCl_2_ with Tween 80 for 20 min [20]. Yan et al. (2020) proposed a multi-step protocol involving 70% ethanol for 1 min, 0.5% NaClO with Tween 80 for 5 min, and 0.1% HgCl_2_ with Tween 80 for 4 min [21]. These procedures are relatively complex and commonly use mercuric chloride. Because mercury compounds are highly toxic and pose environmental risks, their use requires strict handling and waste management [22]. Here, we aimed to simplify surface sterilization by developing an optimized regeneration protocol that reduces contamination, minimizes tissue damage, and eliminates the need for mercury-based disinfectants.

Despite decades of work on *B. papyrifera* tissue culture, several challenges remain, including low transformation efficiency and a limited range of target genes that can be routinely transformed [23]. In particular, published studies vary widely in the types, concentrations, and combinations of plant growth regulators used at each stage of regeneration, and systematic comparisons are scarce. Therefore, we optimized media compositions and plant growth regulator ratios across the key stages of regeneration and transformation, aiming to establish a stable, efficient regeneration protocol that can serve as a standardized technical reference for *B*. *papyrifera*.

With the availability of genome resources for *B. papyrifera*, efficient genetic transformation has become an essential prerequisite for functional genomics. It is also needed to improve traits related to ecological restoration and the production of valuable secondary metabolites. Comparative genomics indicated an expansion of gene families involved in flavonoid biosynthesis in *B. papyrifera* [24]. This suggests that flavonoid metabolism may contribute to its tolerance of diverse abiotic and biotic stresses. Therefore, functional characterization of flavonoid-related gene families in *B*. *papyrifera* is essential for understanding stress tolerance mechanisms. Here, we aimed to generate *B*. *papyrifera* plants overexpressing the chalcone synthase gene *BpCHS8* (Chalcone Synthase 8 of *B*. *papyrifera*, *BpCHS8*) via genetic transformation. This will provide valuable materials for investigating the role of chalcone-mediated flavonoid metabolism in stress resistance. Agrobacterium-mediated transformation has been used to introduce *BpTT2* into *B. papyrifera*, producing lines with up to a 7.22-fold increase in transcript abundance [25]. Similarly, overexpression of *AtNHX5* enhanced salt and drought tolerance in Hybrid *B. papyrifera* [26]. However, existing transformation workflows remain incomplete and often depend heavily on previously established regeneration systems. Accordingly, we aim to establish and optimize a leaf-based regeneration and transformation system using explants collected from greenhouse-grown plants. This will provide a practical platform for downstream gene functional studies.

## 2. Materials and Methods

### 2.1. Plant Materials

The Hybrid *Broussonetia papyrifera* plants used in this study were provided by Prof. Shihua SHEN (Chinese Academy of Plant Sciences) and maintained in the greenhouse of the College of Forestry and Landscape Architecture, South China Agricultural University (Guangzhou, China).

### 2.2. Culture Media Optimization for Each Stage of B. papyrifera Tissue Culture

#### 2.2.1. Evaluation of Sterilization Methods for Hybrid *B. papyrifera* Leaf Explants

Healthy, disease-free young leaves from Hybrid *B. papyrifera* were collected as explants. Leaves were washed with a mild detergent, gently brushed, and flushed with flowing water for 10–15 min. For surface sterilization, explants were treated with 0.6% sodium hypochlorite (NaClO, Shanghai Macklin Biochemical Co., Ltd., Shanghai, China), 0.3% mercuric chloride (HgCl_2_, Shanghai Aladdin Biochemical Technology Co., Ltd., Shanghai, China), or 10% hydrogen peroxide (H_2_O_2_, Guangzhou Chemical Reagent Factory, Guangzhou, China) for 8 or 15 min (Table 1). After surface sterilization, explants were rinsed 5–6 times with sterile water and blotted dry on sterile filter paper. Leaf margins and petioles were removed with sterile scissors or a scalpel, and leaves were cut into 0.5 cm × 0.5 cm discs. Discs were placed on Murashige and Skoog salts (MS, PhytoTechnology Laboratories, Lenexa, KS, USA) solid medium with 6 explants per bottle, 5 bottles per treatment, and 3 biological replicates. The cultures were then incubated under controlled light conditions. After 7 days, contamination rate, browning rate, and survival rate were recorded.Contamination rate (%) = (Number of contaminated explants/Total number of explants) × 100Browning rate (%) = (Number of browned explants/Total number of explants) × 100Survival rate (%) = (Number of surviving explants/Total number of explants) × 100

#### 2.2.2. Screening of Callus Induction Medium

Leaf discs were placed on callus induction medium (CIM). The basal medium was Murashige and Skoog (MS) salts supplemented with 30 g·L^−1^ sucrose (Shanghai Macklin Biochemical Co., Ltd., Shanghai, China), 0.7 g·L^−1^ 2-(N-morpholino) ethanesulfonic acid (MES, Shanghai Aladdin Biochemical Technology Co., Ltd., Shanghai, China), and 8 g·L^−1^ agar (Biofroxx, Einhausen, Germany) (pH 5.8–6.0). A two-factor orthogonal design was used to test the effects of 6-benzyladenine (6-BA; 1.00, 1.50, and 2.00 mg·L^−1^) (Shanghai Macklin Biochemical Co., Ltd., Shanghai, China) and indole-3-butyric acid (IBA; 0.05, 0.10, and 0.15 mg·L^−1^) (Shanghai Macklin Biochemical Co., Ltd., Shanghai, China) on callus induction (Table 2). For each treatment, 40 explants from independent biological sources were cultured in the dark at 25 ± 2 °C. Callus morphology was monitored regularly. After 45 days, callus induction rate and callus quality were evaluated to identify the optimal CIM formulation.Callus induction rate (%) = (Number of explants producing callus/Total number of explants) × 100

#### 2.2.3. Screening of Shoot Induction Medium

Calli induced from Hybrid *B. papyrifera* leaf explants were transferred to the shoot induction medium (SIM). The basal medium was MS-supplemented with 30 g·L^−1^ sucrose, 0.7 g·L^−1^ MES, and 8 g·L^−1^ agar (pH 5.8–6.0). Plant growth regulators 6-BA, IBA, and gibberellic acid (GA_3_, Shanghai Macklin Biochemical Co., Ltd., Shanghai, China) were tested in an orthogonal design (Table 3) to promote adventitious shoot induction. The IBA concentration was fixed at 0.05 mg·L^−1^, while 6-BA at 1.00, 1.50, and 2.00 mg·L^−1^ and GA_3_ at 0.50, 1.00, and 1.50 mg·L^−1^ were varied. For each treatment, six Petri dishes were used (five calli per dish), with three biological replicates. After 60 days, the adventitious shoot induction rate and shoot conversion rate were calculated.Adventitious shoot induction rate (%) = (Number of explants producing shoots/Total number of explants) × 100

#### 2.2.4. Screening of Shoot Multiplication Medium

Adventitious shoots approximately 1.0 cm in length were transferred to the shoot multiplication medium (SMM). The basal medium was MS-supplemented with 30 g·L^−1^ sucrose, 0.7 g·L^−1^ MES, and 8 g·L^−1^ agar (pH 5.8–6.0). An orthogonal design (Table 4) was used to evaluate 6-BA at 0.15, 0.50, and 1.00 mg·L^−1^ and IBA at 0.05, 0.10, and 0.15 mg·L^−1^ on shoot multiplication. For each treatment, four culture bottles were used with five shoots per bottle and three biological replicates. After 30 days, the shoot multiplication coefficient was calculated.Shoot multiplication coefficient = (Number of newly produced shoots/Number of inoculated shoots)

#### 2.2.5. Screening of Rooting Medium

Shoots approximately 1.5 cm in length, excised from multiplicated shoot clusters, were transferred to the rooting medium. The basal medium was 1/2-strength MS-supplemented with 30 g·L^−1^ sucrose and 8 g·L^−1^ agar (pH 5.8–6.0). Four concentrations of IBA (0.05, 0.10, 0.15, and 0.20 mg·L^−1^) were tested (Table 5). For each treatment, twenty culture bottles were used with two shoots per bottle and three biological replicates. After 25 days, the rooting rate was recorded.Rooting rate (%) = (Number of rooted plantlets/Total number of inoculated plantlets) × 100

### 2.3. Optimization of Genetic Transformation System for B. papyrifera

#### 2.3.1. BpCHS8 Gene Cloning and Overexpression Vector Construction

Total RNA was extracted from *B. papyrifera* roots and reverse-transcribed into cDNA. The full-length coding sequence (CDS) of *BpCHS8* (*Broussonetia papyrifera* Chalcone Synthase 8, *BpCHS8*) was amplified from cDNA using gene-specific primers containing restriction enzyme sites for cloning into the pCAMBIA1300-35S-eGFP (ps1300, Catalog No.: V013557, NovoPro Bioscience Inc., Somerville, MA, USA) vector [27] (forward primer with a *Kpn*I site: GGGGGGTACCATGGTGACCGTTGAGGAAGTT; reverse primer with an *Xba*I site: GGGTCTAGATCTAAATAGAAACACTGTGGAGCAC). PCR amplification was performed under the following conditions: an initial denaturation at 94 °C for 3 min; 35 cycles of 94 °C for 30 s, 59 °C for 30 s, and 72 °C for 90 s; followed by a final extension at 72 °C for 10 min. The amplified products were double-digested with Takara QuickCut™ *Xba* I (TaKaRa Bio Inc., Shiga, Japan) and Takara QuickCut™ *Kpn*I (TaKaRa Bio Inc., Shiga, Japan) and ligated into the corresponding vector, ps1300, at 16 °C, using T4 DNA Ligase (Vazyme Biotech Co., Ltd., Nanjing, China). The ligation mixture was transformed into *Escherichia coli* DH5α (Shanghai Weidi Biotechnology Co., Ltd., Shanghai, China) competent cells. Positive colonies were screened by colony PCR, and recombinant plasmids were further confirmed by PCR amplification and restriction enzyme digestion. The insert sequence was finally verified by Sanger sequencing (Sangon Biotech Co., Ltd., Shanghai, China).

#### 2.3.2. Preparation of *A. tumefaciens* Suspension

The recombinant vector was introduced into *A*. *tumefaciens* strain EHA105 (Shanghai Weidi Biotechnology Co., Ltd., Shanghai, China) using a liquid nitrogen freeze–thaw method [28], and positive clones were stored at −80 °C. For activation, a single colony was cultured on LB medium containing 50 mg·L^−1^ kanamycin (Shanghai yuanye Bio-Technology Co., Ltd., Shanghai, China) at 28 °C for 2 days. A positive colony was then inoculated into LB liquid medium containing 50 mg·L^−1^ rifampicin (Shanghai yuanye Bio-Technology Co., Ltd., Shanghai, China) and 50 mg·L^−1^ kanamycin and grown overnight at 28 °C with shaking (180–200 rpm). Next, 2 mL of the overnight culture was transferred into 25 mL fresh LB liquid medium containing rifampicin (50 mg·L^−1^) and kanamycin (50 mg·L^−1^), and incubated for 2–3 h at 28 °C with shaking (180–200 rpm) until OD_600_ reached 0.2–0.3. Cells were collected by centrifugation, resuspended in MS liquid medium containing 200 μmol acetylsyringone (AS, Shanghai Aladdin Biochemical Technology Co., Ltd., Shanghai, China), and pre-induced for 30–60 min at 28 °C with shaking (180–200 rpm) before infection.

#### 2.3.3. Leaf-Disc Method for Infection and Transformation

Sterile leaves were cut into 0.5 cm × 0.5 cm discs using sterile scissors. Leaf discs were immersed in the prepared *A*. *tumefaciens* suspension for 20 min with gentle agitation. Excess bacterial suspension was removed by blotting on sterile filter paper.

Leaf discs were placed on MS medium supplemented with 200 μmol AS and overlaid with sterile filter paper for co-cultivation (2 days). After co-cultivation, explants were washed twice for 5 min in MS liquid medium containing 250 mg·L^−1^ cefotaxime (Shanghai yuanye Bio-Technology Co., Ltd., Shanghai, China) and 200 mg·L^−1^ Timentin (Shanghai Macklin Biochemical Co., Ltd., Shanghai, China) with gentle shaking, followed by three rinses with sterile water. Explants were then transferred to callus induction medium.

Hygromycin (Hyg, Shanghai Macklin Biochemical Co., Ltd., Shanghai, China), an antibiotic used for selection of genetically modified cells, was used for selection because the vector carries a hygromycin-resistance marker. To determine the appropriate selection pressure, Hyg was tested at 0, 2.0, 3.5, 5.0, 6.5, and 8.0 on MS-based medium supplemented with the optimized plant growth regulator combination and 250 mg·L^−1^ cefotaxime. Post-co-cultivation leaf explants lacking genetic transformation with the ps1300 vector were cultured on selection medium, with five bottles used per concentration, six explants per bottle, and three biological replicates. The survival rate of explants was recorded for each group, with the final value for each experimental group determined as the arithmetic mean of three independent biological replicates.

### 2.4. Identification of Positive Transgenic Lines

#### 2.4.1. Method for Evolutionary Analysis of BpCHS8

The amino acid sequence of *BpCHS8* was obtained from the *B*. *papyrifera* genome [24]. Homologous *CHS* sequences from *Populus trichocarpa* and *Morus notabilis* were retrieved from Phytozome (*Populus trichocarpa* v4.1, https://phytozome-next.jgi.doe.gov/) and the *Morus* Genome Database *(M. notabilis* C.K. Schneid, http://morus.swu.edu.cn/morusdb/, accessed on 24 December 2025), respectively. Amino acid sequences were aligned with the ClustalW tool integrated in MEGA11 using default parameters, and a neighbour-joining phylogenetic tree was constructed in the same software with node support assessed via 1000 bootstrap replicates.

#### 2.4.2. DNA-Level Identification of Transformants

Putative *BpCHS8*::ps1300 transformants were first screened for enhanced green fluorescent protein (eGFP) fluorescence using an upright fluorescence microscope (Nikon Eclipse Ni-U; excitation wavelength: 465–495 nm). Genomic DNA was then extracted using the E.Z.N.A.^®^ HP Plant DNA Mini Kit (Omega Bio-tek, Norcross, GA, USA) according to the manufacturer’s instructions. PCR amplification was performed with primers *BpCHS8*+eGFP-F (CCAAGGACCTTGCAGAGAACA) and *BpCHS8*+eGFP-R (GGCTGTTGTAGTTGTACTCCAGCT). The *BpCHS8*::ps1300 plasmid served as a positive control, while genomic DNA from non-transgenic plants and from plants transformed with the empty ps1300 vector served as negative controls. PCR products were analyzed by agarose gel electrophoresis.

#### 2.4.3. qRT-PCR Analysis

Total RNA was extracted from uniformly growing *BpCHS8*::ps1300 transgenic plantlets and non-transgenic plantlets using the E.Z.N.A.^®^ Plant RNA Kit (Omega Bio-tek, Norcross, United States) according to the manufacturer’s instructions. First-strand cDNA was synthesized using the TaKaRa PrimeScript^TM^ RT Reagent Kit (TaKaRa Bio Inc., Shiga, Japan). Gene-specific primers were used for the reference gene *BpDOUB* (*Broussonetia papyrifera* DOUBle-stranded RNA-binding protein, *BpDOUB*) and the target gene *BpCHS8*. The primer sequences were as follows: *BpDOUB*-F: (CCTGATCTTCGCCGGAAAACA) and *BpDOUB*-R: (TGGAGAGGGTTGAAGAGAGCT) for the reference gene; and *BpCHS8*-rt-F: (CCGTCAAACGTCTGATGATG) and *BpCHS8*-rt-R: (AATAAGGCTTGACCCACCAA) for the target gene. Quantitative PCR was performed using the Taq Pro Universal SYBR qPCR Master Mix (Vazyme Biotech Co., Ltd., Nanjing, China) in a 20 μL reaction containing 10 μL 2× master mix, 0.4 μL forward primer (10 μM), 0.4 μL reverse primer (10 μM), 2 μL diluted cDNA template, and 7.2 μL nuclease-free water. The thermal cycling programme was as follows: 95 °C for 30 s; 40 cycles of 95 °C for 5 s; and 60 °C for 30 s; and it was followed by melt-curve analysis to confirm primer specificity. Four technical replicates were run for each sample. Relative expression levels were calculated using the 2^−ΔΔCt^ method, with *BpDOUB* as the internal reference.

### 2.5. Statistical Analyses

All experimental data were collated and summarized using Microsoft Excel 2016, and statistical analyses were performed with IBM SPSS Statistics 23 software. Two-way analysis of variance (two-way ANOVA) was used to evaluate the main effects of factors and their interactions on relevant indicators for the following analyses: comparison of the disinfection effects on explant under different treatments (Table 1), effects of different plant growth regulator concentration combinations on leaf callus induction (Table 2), effects of plant growth regulators on adventitious shoot regeneration rate from Hybrid *B*. *papyrifera* leaves (Table 3), and effects of plant growth regulators on shoot multiplication coefficient of Hybrid *B. papyrifera* (Table 4). One-way analysis of variance (one-way ANOVA) was used to analyze the effects of plant growth regulators on the rooting rate of Hybrid *B*. *papyrifera* (Table 5). After ANOVA, Duncan’s multiple range test was used to determine the significance of differences between groups, with the significance level set at *p* < 0.05. All data are presented as “mean ± standard error (mean ± SE)”. Graphs were generated using GraphPad Prism 9.5 software, and bar charts were used to display the mean values and standard errors of indicators in each group for intuitive visualization of intergroup differences.

## 3. Results

### 3.1. Optimization of In Vitro Regeneration of Hybrid B. papyrifera

#### 3.1.1. Optimization of Leaf Explant Sterilization

To identify an effective and non-destructive sterilization procedure, we compared three disinfectants (10% H_2_O_2_, 0.3% HgCl_2_, and 0.6% NaClO) at two treatment durations (8 and 15 min). Both disinfectant type and exposure time significantly affected contamination, browning, and explant survival (Table 1). For H_2_O_2_, extending the treatment from 8 to 15 min reduced contamination from 39.43% to 26.27% but markedly increased browning to 58.73%. HgCl_2_ caused substantial contamination and browning at both time points, with contamination ranging from 37.78 to 40.83%, and browning from 41.77 to 42.22%. In contrast, NaClO provided the best overall balance. An 8 min NaClO treatment produced the lowest browning at 31.70%, with acceptable contamination at 35.84% and the highest survival at 33.60% among treatments, whereas a 15 min treatment further reduced contamination to 28.80% but increased browning to 50.53%. Considering both cleanliness and tissue viability, 0.6% NaClO for 8 min was selected for subsequent experiments.

#### 3.1.2. Callus Induction from Leaf Explants

Leaf discs of 0.5 cm × 0.5 cm were cultured on CIM containing different combinations of 6-BA and IBA. Callus formation and morphology were evaluated after 45 days (Figure 1; Table 2). Media containing 1.0 mg·L^−1^ 6-BA combined with IBA resulted in pronounced browning and poor-quality callus (Figure 1a–c). When 6-BA was 1.5 mg·L^−1^, callus induction rates were high at 90.0–97.5%, and calli were dense and yellow-green (Figure 1d–f). Increasing 6-BA to 2.0 mg·L^−1^ reduced induction rates to 75.0–90.0% and produced yellow-brown callus (Figure 1g–i). Among the nine combinations, the medium supplemented with 1.5 mg·L^−1^ 6-BA and 0.15 mg·L^−1^ IBA not only achieved the highest callus induction rate at 97.5% but also produced high-quality calli, thus being selected as the optimal CIM formulation.

#### 3.1.3. Adventitious Shoot Induction

Induced calli were transferred to SIM to promote shoot induction. Based on preliminary tests, IBA was fixed at 0.05 mg·L^−1^, while 6-BA and GA_3_ were varied (Table 3). Both the adventitious shoot induction rate and the shoot conversion rate differed substantially among treatments (Figure 2, Table 3). At 1.0 mg·L^−1^ 6-BA + 0.05 mg·L^−1^ IBA, induction remained 50–65% as GA_3_ increased from 0.5 to 1.5 mg·L^−1^; however, shoot conversion was low or absent at 0–12.11%. At 1.5 mg·L^−1^ 6-BA + 0.05 mg·L^−1^ IBA, increasing GA_3_ from 0.5 to 1.5 mg·L^−1^ reduced induction from 90.00% to 82.22% and decreased shoot conversion from 83.89% to 60.57%. At 2.0 mg·L^−1^ 6-BA + 0.05 mg·L^−1^ IBA, the adventitious shoot induction rate and shoot conversion rate were generally lower than those recorded for the 1.5 mg·L^−1^ 6-BA + 0.05 mg·L^−1^ IBA combination across most GA_3_ concentrations tested. Overall, SIM4 (MS + 1.5 mg·L^−1^ 6-BA + 0.05 mg·L^−1^ IBA + 0.5 mg·L^−1^ GA_3_) provided the most efficient production of normal shoots and was selected for subsequent experiments.

#### 3.1.4. Shoot Multiplication

To optimize shoot multiplication, regenerated shoots were transferred to SMM containing different concentrations of 6-BA and IBA (Table 4). The highest multiplication coefficient at 5.70 was obtained in SMM1 (0.15 mg·L^−1^ 6-BA + 0.05 mg·L^−1^ IBA). At the same 6-BA concentration of 0.15 mg·L^−1^, increasing IBA to 0.10 or 0.15 mg·L^−1^ resulted in slightly lower multiplication coefficient at 5.27 and 5.47. Increasing 6-BA to 0.50 mg·L^−1^ reduced the multiplication coefficient to a range of 2.33–4.13, and 1.00 mg·L^−1^ 6-BA markedly inhibited shoot growth to a range of 0.94–2.07. Therefore, MS + 0.15 mg·L^−1^ 6-BA + 0.05 mg·L^−1^ IBA was selected as the optimal SMM formulation.

#### 3.1.5. Rooting of Regenerated Shoots

Shoots were transferred to 1/2 MS rooting media containing different IBA concentrations (Table 5). Rooting occurred in all treatments, with rooting rates ranging from 64.17% to 89.17%. The highest rooting rate at 89.17% was obtained at 0.05 mg·L^−1^ IBA. Rooting declined slightly at 0.10 mg·L^−1^ IBA and decreased substantially at ≥0.15 mg·L^−1^ IBA, with the poorest rooting observed at 0.20 mg·L^−1^ IBA. Thus, the optimal rooting medium was 1/2 MS + 0.05 mg·L^−1^ IBA.

### 3.2. Optimization of Genetic Transformation for Hybrid B. papyrifera

#### 3.2.1. Determination of Hygromycin Selection Concentration

To determine an appropriate hygromycin concentration for selection, leaf explants were cultured on differentiation medium containing 0–8.0 mg·L^−1^ hygromycin for 30 days. Explants were highly sensitive to hygromycin, and regeneration decreased as hygromycin concentration increased (Figure 3). At 3.5 mg·L^−1^, only approximately 5% of explants survived. At 5.0 mg·L^−1^, explant growth and differentiation were completely inhibited. Therefore, 5 mg·L^−1^ hygromycin was defined as the lethal critical concentration for non-transformed leaf explants and was used as a benchmark for selection stringency.

#### 3.2.2. Evolutionary Analysis Results of BpCHS8

Chalcone synthase (CHS) catalyzes the first committed step of the flavonoid biosynthetic pathway and is widely considered a rate-limiting enzyme [29]. It catalyzes the condensation of p-coumaroyl-CoA with malonyl-CoA to produce chalcone, the precursor for multiple branches of flavonoid biosynthesis. *CHS* genes are also responsive to abiotic stresses such as salinity, alkaline stress, and heavy metals [30].

To examine the evolutionary relationship of *BpCHS8* with homologs from other species, we constructed a phylogenetic tree based on amino acid sequences from *B. papyrifera*, *Morus notabilis*, and *Populus trichocarpa* (Figure 4). *BpCHS8* clustered within the Moraceae clade together with *MnCHS1* and *MnCHS2* from *M. notabilis*, consistent with their close taxonomic relationship. In contrast, *P. trichocarpa* CHS homologs formed a separate sister clade, indicating greater evolutionary distance between Moraceae and Salicaceae. This pattern suggests that *CHS* genes are more conserved within Moraceae and implies that *BpCHS8* may share functions with Moraceae CHS homologs.

### 3.3. Genetic Transformation and Molecular Identification of BpCHS8::ps1300 Lines

The overall workflow for *B. papyrifera* transformation is summarized in Figure 5 and Figure 6. After infection with *A. tumefaciens* and 2 days of co-cultivation, explants were washed with an antibiotic solution to eliminate bacteria and were placed on selection medium to induce callus. eGFP fluorescence was evaluated at approximately 28 days, after which fluorescent calli were transferred to shoot induction medium. Resistant shoots typically appeared after approximately 60 days and were excised and transferred to shoot multiplication medium. After 20–30 days of multiplication, resistant shoots were transferred to rooting medium, and rooted plantlets were obtained after approximately 28 days. Molecular confirmation (PCR and qRT-PCR) was performed after approximately 150 days from infection.

#### 3.3.1. Fluorescence Screening

Putative transformants were screened by detecting eGFP fluorescence in callus tissue using a fluorescence microscope. Strong green fluorescence in calli indicated successful expression of the eGFP gene and served as an initial indicator of transformation (Figure 7).

#### 3.3.2. PCR Confirmation of Transgene Integration

Genomic DNA was extracted from roots of hygromycin-resistant putative transformants and used as a template for PCR amplification of the *BpCHS8* transgene. A specific amplicon was detected in six independent transgenic lines and the positive control *BpCHS8*::ps1300 plasmid, while no band was detected in the wild-type control (Figure 8A). These results confirm stable integration of *BpCHS8* into the genome of the transgenic lines.

#### 3.3.3. qRT-PCR Analysis of BpCHS8 Expression

To evaluate transgene expression, qRT-PCR was performed using root samples from six transgenic lines and the wild-type line. Compared with wild-type, *BpCHS8* transcript abundance was significantly increased in several lines (Figure 8B). Overall, the *BpCHS8* transcript in transgenic lines was 6.32–43.36-fold higher than in wild-type, demonstrating successful overexpression of *BpCHS8* in *B. papyrifera*.

## 4. Discussion

A number of *B. papyrifera* regeneration systems have been reported, including protocols based on petioles, leaves, shoot tips, and axillary buds [31,32,33,34,35,36]. Among these studies, Yu et al. (2006) established a tissue regeneration method using petioles and leaves from Hybrid *B. papyrifera* as explants [31]. Li et al. (2007) were the first to provide a comprehensive and systematic description of surface sterilization protocols for vegetative organs, including shoot tips, axillary buds, and leaves of *B. kazinoki*, thereby successfully establishing a tissue regeneration system for this species [19]. Wei (2010) employed leaf explants from in vitro-grown Hybrid *B. papyrifera* to investigate the effects of different plant growth regulators on adventitious shoot induction [18]. Subsequently, Li (2015) developed a large-scale somatic embryogenesis system using organ-derived tissues, resulting in the production of artificial seeds with a germination rate of up to 33% [32]. Following this, Zhang (2016) established a stable cell suspension culture system for *B. papyrifera* by constructing a dynamic kinetic model of suspension cell growth [33]. More recently, Tian et al. (2019) investigated surface sterilization strategies using stem segments of Hybrid *B. papyrifera* as explants and reported a maximum explant survival rate of 44% [34]. Additionally, Lin et al. (2023) induced polyploidy in *B. papyrifera* leaves, calli, and seeds using colchicine treatment, yielding tetraploid plants characterized by increased stem diameter and enlarged stomata [35]. In another advancement, Liu (2021) established axenic *B. papyrifera* seedlings from seeds through surface sterilization with ethanol and mercuric chloride [36]. Nevertheless, many studies still rely on multi-step sterilization procedures involving ethanol, sodium hypochlorite, and mercuric chloride, which increases labour and environmental burden. To address this, in this study, we simplified sterilization by using a single 0.6% NaClO treatment for 8 min after thorough washing. This approach reduced tissue damage, achieved a high survival rate, and reduced contamination, while eliminating the need for HgCl_2_ and thereby reducing environmental and safety concerns. Compared with previously reported protocols, this simplified procedure is easier to implement for routine transformation experiments using leaf explants.

Plant growth regulators are central to successful regeneration. In general, higher cytokinin-to-auxin ratios promote shoot formation, whereas higher auxin levels favour rooting [37]. Accordingly, we optimized hormone combinations for each stage of regeneration. The highest callus induction rate reached 97.5% on MS medium supplemented with 1.5 mg·L^−1^ 6-BA and 0.15 mg·L^−1^ IBA. Efficient shoot induction and conversion were achieved on MS + 1.5 mg·L^−1^ 6-BA + 0.05 mg·L^−1^ IBA + 0.5 mg·L^−1^ GA_3_. For shoot multiplication, MS + 0.15 mg·L^−1^ 6-BA + 0.05 mg·L^−1^ IBA produced the highest multiplication coefficient of 5.70, and rooting was optimal on 1/2 MS medium supplemented with 0.05 mg·L^−1^ IBA, achieving a rooting rate of 89.17%. Collectively, these optimized hormone regimes improved both regeneration efficiency and plantlet quality.

For genetic transformation, we used a leaf-disc method similar to that previously reported for *B. papyrifera* [38], but refined the co-cultivation step by placing infected explants on co-cultivation medium overlaid with sterile filter paper. This modification reduced bacterial overgrowth and improved explant viability. In addition, cefotaxime and Timentin were included after co-cultivation to suppress *A. tumefaciens* without compromising regeneration. Finally, long-term selection with hygromycin facilitated the recovery and identification of stable transgenic lines in this study.

CHS genes play key roles in flavonoid biosynthesis. In *Glycine max*, *GmCHS8* expression influences seed isoflavone levels [39], and in *Vaccinium corymbosum*, transient overexpression of *VcCHS8* enhances anthocyanin accumulation and upregulates flavonoid pathway genes [40]. Because *B. papyrifera* is rich in flavonoids, characterizing and manipulating *BpCHS8* may support metabolic engineering aimed at increasing high-value secondary metabolites. In this context, the *BpCHS8*-overexpressing *B. papyrifera* lines generated in this study provide a practical genetic resource for evaluating the contribution of *BpCHS8* to flavonoid accumulation and stress-related traits. Moreover, as genome resources and gene-editing tools expand, the regeneration and transformation system described here will support future functional studies and genetic improvement efforts in *B. papyrifera*.

## 5. Conclusions

In this study, we established an efficient and reproducible regeneration and transformation system for Hybrid *B. papyrifera* using leaf explants. A simplified sterilization protocol with 0.6% NaClO for 8 min achieved an explant survival rate of 33.60% and reduced contamination to 35.84% while avoiding mercury-based sterilants. Optimal media and plant growth regulator combinations were identified for each stage: callus induction (MS + 1.5 mg·L^−1^ 6-BA + 0.15 mg·L^−1^ IBA; 97.5% induction), shoot induction (MS + 1.5 mg·L^−1^ 6-BA + 0.05 mg·L^−1^ IBA + 0.5 mg·L^−1^ GA_3_), shoot multiplication (MS + 0.15 mg·L^−1^ 6-BA + 0.05 mg·L^−1^ IBA; multiplication coefficient 5.70), and rooting (1/2 MS + 0.05 mg·L^−1^ IBA; 89.17% rooting). For transformation, the lethal hygromycin concentration for leaf explants was 5 mg·L^−1^, and the co-cultivation step was improved by using a filter paper overlay and antibiotic suppression with 250 mg·L^−1^ cefotaxime and 200 mg·L^−1^ Timentin to limit *A. tumefaciens* overgrowth. Using this workflow, we obtained PCR- and qRT-PCR-confirmed transgenic lines. Overall, this optimized system provides a robust platform for functional genomics and molecular breeding of *B. papyrifera*.

## Figures and Tables

**Figure 1 life-16-00078-f001:**
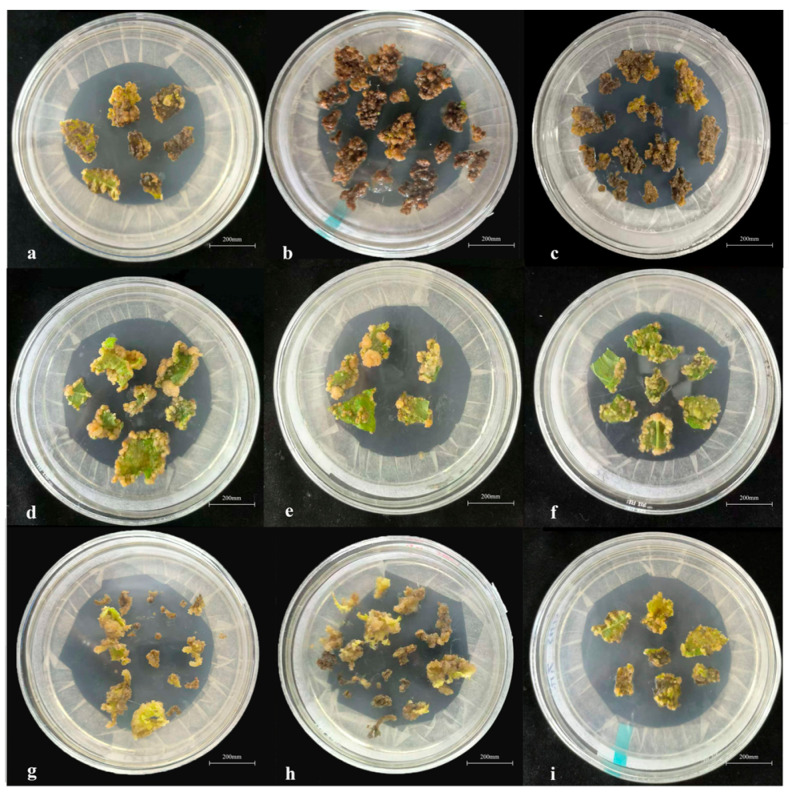
Callus induction of Hybrid *B. papyrifera* under different hormone treatments. Note: (**a**) 1 mg·L^−1^ 6-BA + 0.05 mg·L^−1^ IBA; (**b**) 1 mg·L^−1^ 6-BA + 0.1 mg·L^−1^ IBA; (**c**) 1 mg·L^−1^ 6-BA + 0.15 mg·L^−1^ IBA; (**d**) 1.5 mg·L^−1^ 6-BA + 0.05 mg·L^−1^ IBA; (**e**) 1.5 mg·L^−1^ 6-BA + 0.1 mg·L^−1^ IBA; (**f**) 1.5 mg·L^−1^ 6-BA + 0.15 mg·L^−1^ IBA; (**g**) 2.0 mg·L^−1^ 6-BA + 0.05 mg·L^−1^ IBA; (**h**) 2.0 mg·L^−1^ 6-BA + 0.1 mg·L^−1^ IBA; and (**i**) 2.0 mg·L^−1^ 6-BA + 0.15 mg·L^−1^ IBA.

**Figure 2 life-16-00078-f002:**
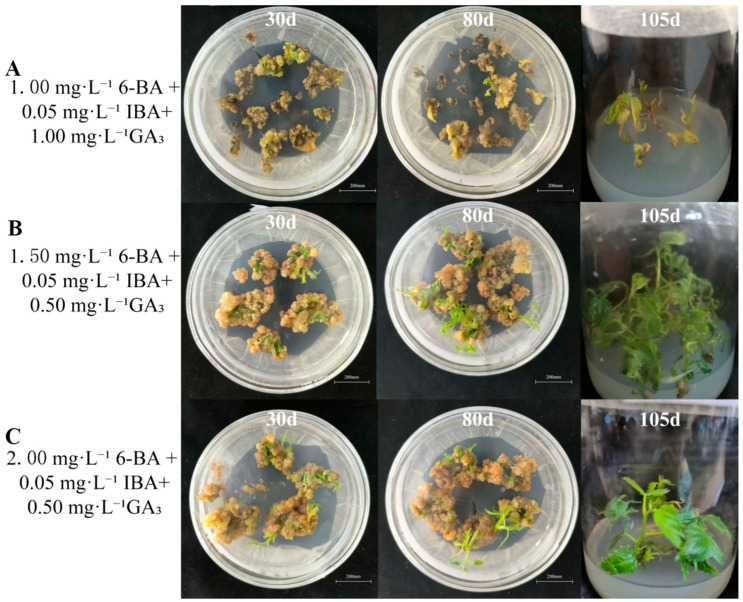
Regeneration of adventitious shoots from leaf-derived calli of Hybrid *B. papyrifera* under different plant growth regulator treatments. Note: (**A**) Growth status of adventitious shoots regenerated from leaf-derived calli on SIM2 (1.00 mg·L^−1^ 6-BA + 0.05 mg·L^−1^ IBA + 1.00 mg·L^−1^ GA_3_); (**B**) growth status of adventitious shoots regenerated from leaf-derived calli on SIM4 (1.50 mg·L^−1^ 6-BA + 0.05 mg·L^−1^ IBA + 0.50 mg·L^−1^ GA_3_); and (**C**) growth status of adventitious shoots regenerated from leaf-derived calli on SIM7 (2.00 mg·L^−1^ 6-BA + 0.05 mg·L^−1^ IBA + 0.50 mg·L^−1^ GA_3_).

**Figure 3 life-16-00078-f003:**
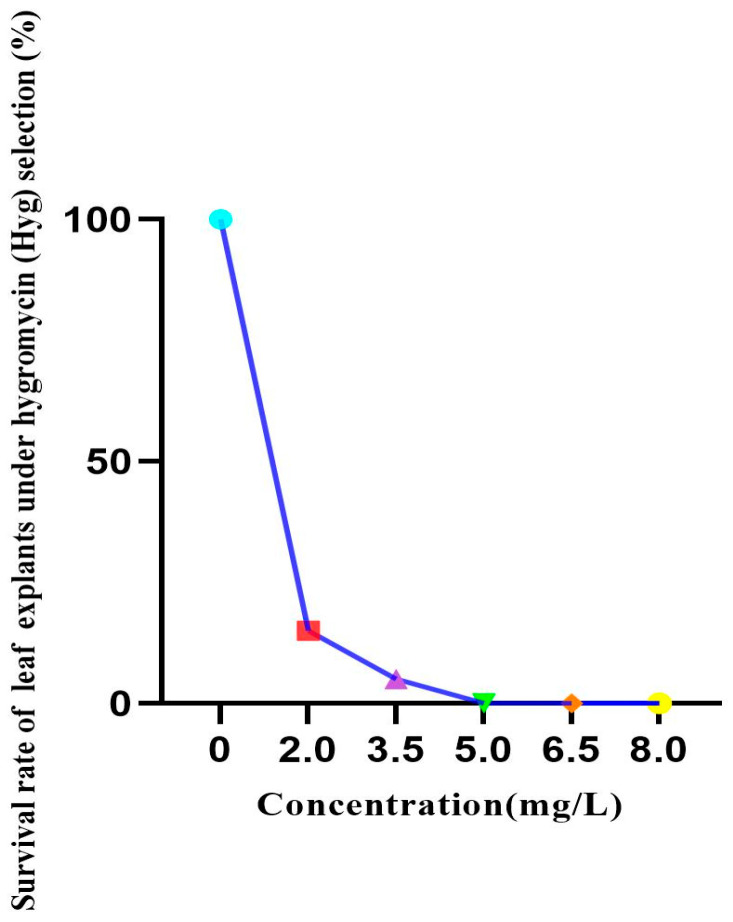
Survival rate of leaf explants under hygromycin (Hyg) selection.

**Figure 4 life-16-00078-f004:**
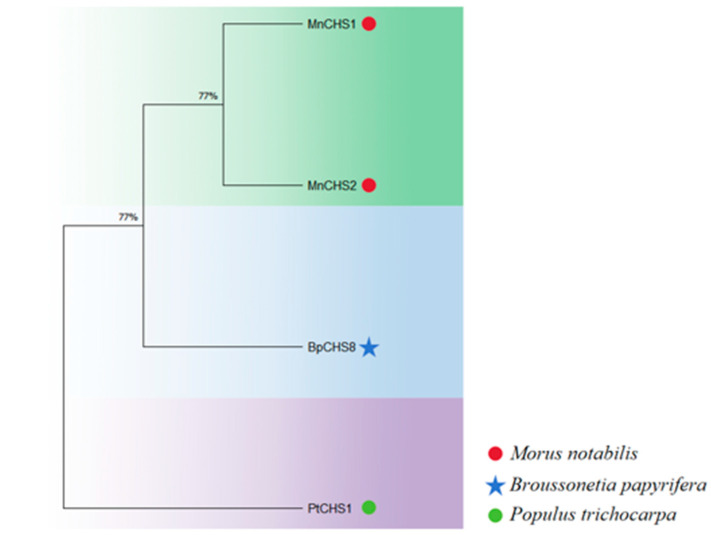
Neighbour-joining phylogenetic tree of *BpCHS8* and homologous CHS proteins.

**Figure 5 life-16-00078-f005:**
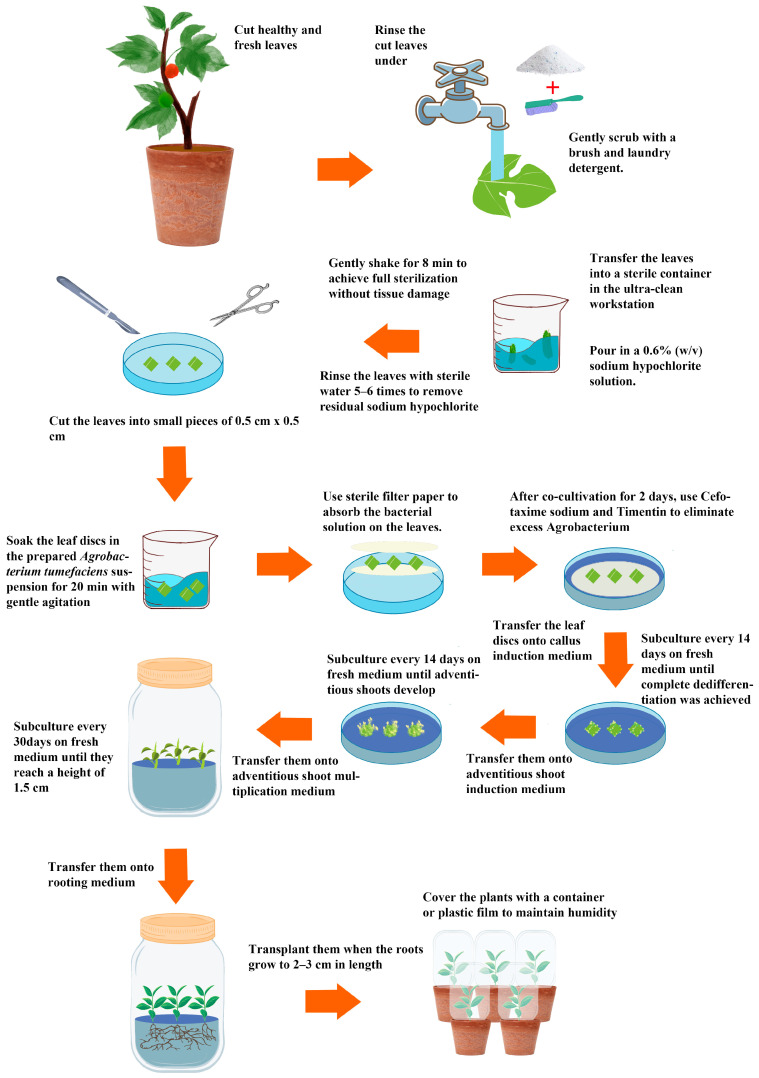
Schematic overview of the Hybrid *B. papyrifera* transformation procedure.

**Figure 6 life-16-00078-f006:**
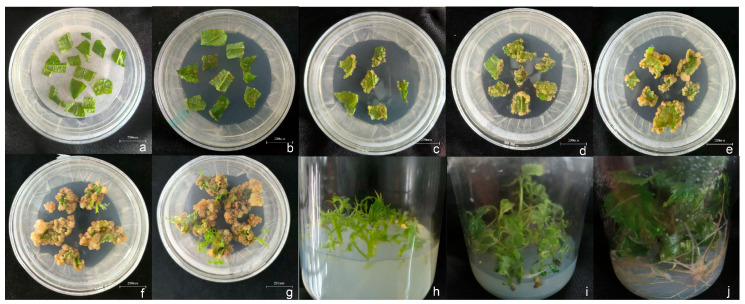
Representative stages of Hybrid *B. papyrifera* genetic transformation and regeneration. Note: (**a**) infection of leaf discs with *A. tumefaciens*; (**b**–**e**) callus induction and selection from leaf explants; (**f**–**h**) shoot induction from callus tissue; (**i**) shoot multiplication; and (**j**) rooting of regenerated shoots.

**Figure 7 life-16-00078-f007:**
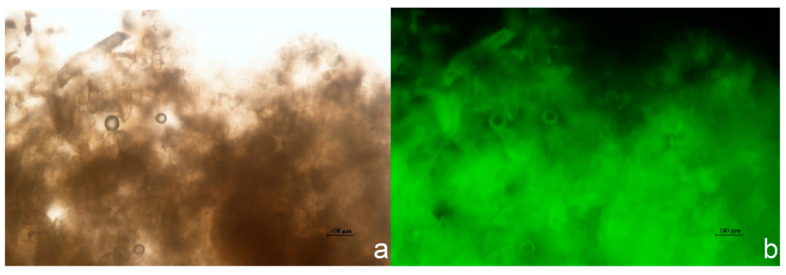
eGFP fluorescence in transgenic callus tissue of *B. papyrifera*. Note: (**a**) bright-field image; and (**b**) fluorescence image.

**Figure 8 life-16-00078-f008:**
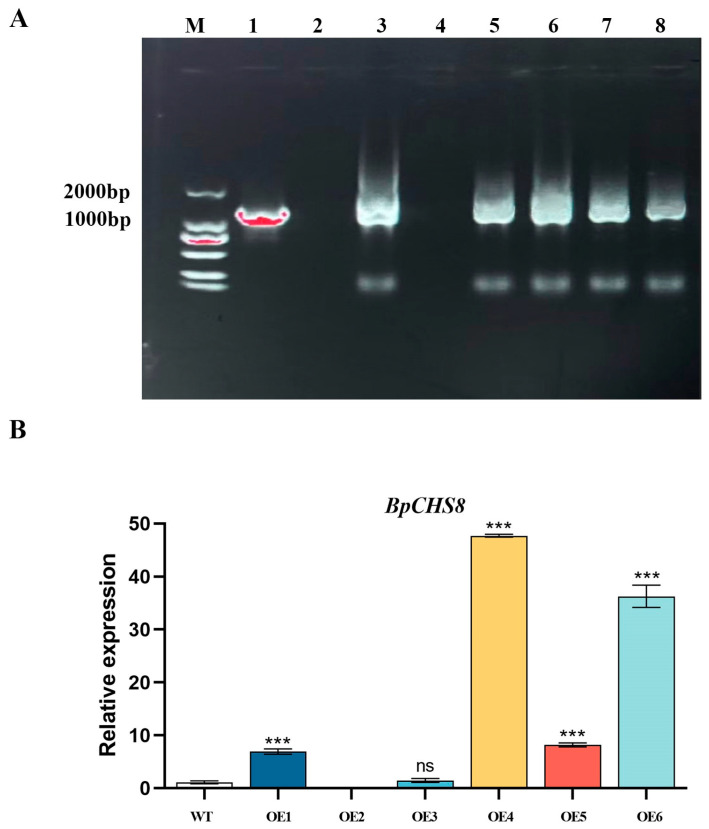
Molecular identification of transgenic lines. Note: (**A**) PCR confirmation of transgenic plants. M: DNA marker; 1: *BpCHS8*::ps1300 plasmid (positive control); 2: wild-type; and 3–8: transgenic lines (OE1–OE6). (**B**) Relative expression of *BpCHS8* in transgenic lines and wild-type determined by qRT-PCR. Expression values were normalized to *BpDOUB*, with the wild-type set to 1 as the reference. Bars represent mean ± SD. “***” indicates a highly significant difference (*p* < 0.001); “ns” indicates no significant difference (*p* > 0.05).

**Table 1 life-16-00078-t001:** Comparison of the disinfection effects on the explant under different treatments.

Disinfection Method	Disinfection Time/min	Number of Explants/Plant	Average Contamination Rate/%	Average Browning Rate/%	Average Explant Survival Rate/%
0.6% NaClO	8	30			
		27			
		29	35.84 ± 11.39 ^a^	31.70 ± 8.72 ^c^	33.6 ± 9.59 ^a^
	15	30			
		29			
		28	28.80 ± 13.34 ^a^	50.53 ± 4.39 ^ab^	23.04 ± 7.31 ^ab^
0.3% HgCl_2_	8	30			
		30			
		30	37.78 ± 13.47 ^a^	42.22 ± 6.94 ^b^	20.00 ± 10.00 ^ab^
	15	30			
		29			
		27	40.83 ± 3.91 ^a^	41.77 ± 5.83 ^b^	17.39 ± 6.57 ^ab^
10% H_2_O_2_	8	30			
		29			
		30	39.43 ± 9.15 ^a^	39.31 ± 1.20 ^b^	21.26 ± 8.18 ^ab^
	15	28			
		30			
		30	26.27 ± 6.08 ^a^	58.73 ± 15.27 ^a^	15.00 ± 9.28 ^b^

Note: Values represented mean ± standard error of three replicates. Different letters within a column indicated significant differences according to Duncan’s multiple range test (*p* < 0.05).

**Table 2 life-16-00078-t002:** Effects of different plant growth regulator concentration combinations on callus induction of the leaf blade.

Treatment	Concentration of 6-BA/(mg·L^−1^)	Concentration of IBA/(mg·L^−1^)	Number of Leaves/Piece	Callus Rate/%	Growth Potential
CIM1	1.00	0.05	40	38/40 = 95	**
CIM2	1.00	0.10	40	38/40 = 95	*
CIM3	1.00	0.15	40	30/40 = 75	*
CIM4	1.50	0.05	40	38/40 = 95	***
CIM5	1.50	0.10	40	36/40 = 90	***
CIM6	1.50	0.15	40	39/40 = 97.5	***
CIM7	2.00	0.05	40	36/40 = 90	***
CIM8	2.00	0.10	40	32/40 = 80	**
CIM9	2.00	0.15	40	30/40 = 75	**

Note: In the table, the asterisks in the “Growth potential” column are used to intuitively characterize the growth status of calli, where “***” indicates the best growth, “**” indicates relatively good growth, and “*” indicates moderate growth.

**Table 3 life-16-00078-t003:** Effects of plant growth regulators on adventitious shoot regeneration rate of Hybrid *B. papyrifera* leaf induction.

Treatment	Concentration of 6-BA/(mg·L^−1^)	Concentration of IBA/(mg·L^−1^)	Concentration of GA_3_/(mg·L^−1^)	Number of Inoculated Explants/Plant	Callus Induction Rate of Adventitious Bud/%	Shoot Conversion Rate %
SIM1	1.00	0.05	0.50	30	50.00 ± 3.30 ^f^	0.33 ± 0.58 ^e^
SIM2	1.00	0.05	1.00	30	65.57 ± 1.96 ^de^	12.11 ± 3.20 ^d^
SIM3	1.00	0.05	1.50	30	60.00 ± 3.33 ^e^	0.00 ± 0.00 ^e^
SIM4	1.50	0.05	0.50	30	90.00 ± 3.33 ^a^	83.89 ± 2.71 ^a^
SIM5	1.50	0.05	1.00	30	86.67 ± 3.34 ^a^	74.29 ± 3.07 ^b^
SIM6	1.50	0.05	1.50	30	82.22 ± 5.09 ^ab^	60.57 ± 7.96 ^c^
SIM7	2.00	0.05	0.50	30	88.89 ± 1.92 ^a^	68.76 ± 1.89 ^bc^
SIM8	2.00	0.05	1.00	30	70.00 ± 3.33 ^cd^	66.62 ± 1.59 ^bc^
SIM9	2.00	0.05	1.50	30	75.55 ± 3.85 ^bc^	63.26 ± 0.66 ^c^

Note: Values represented mean ± standard error of three replicates. Different letters within a column indicated significant differences according to Duncan’s multiple range test (*p* < 0.05). Shoot conversion rate (%) = (Number of normal viable shoots/Total number of differentiated shoots) × 100.

**Table 4 life-16-00078-t004:** Effects of plant growth regulators on shoot multiplication coefficient of Hybrid *B. papyrifera*.

Treatment	Concentration of 6-BA/(mg·L^−1^)	Concentration of IBA/(mg·L^−1^)	Adventitious Shoots/Plant	Shoot Multiplication Coefficient
SMM1	0.15	0.05	20.00	5.70 ± 0.10 ^a^
SMM2	0.15	0.10	20.00	5.27 ± 0.40 ^a^
SMM3	0.15	0.15	20.00	5.47 ± 0.76 ^a^
SMM4	0.50	0.05	20.00	3.03 ± 0.61 ^c^
SMM5	0.50	0.10	20.00	2.33 ± 0.61 ^cd^
SMM6	0.50	0.15	20.00	4.13 ± 0.71 ^b^
SMM7	1.00	0.05	20.00	2.07 ± 0.76 ^cd^
SMM8	1.00	0.10	20.00	1.73 ± 0.71 ^ef^
SMM9	1.00	0.15	20.00	0.93 ± 0.61 ^f^

Note: Values represented mean ± standard error of three replicates. Different letters within a column indicated significant differences according to Duncan’s multiple range test (*p* < 0.05).

**Table 5 life-16-00078-t005:** Effects of plant growth regulators on rooting rate of Hybrid *B. papyrifera*.

Treatment	Concentration of IBA/(mg·L^−1^)	Number of Leaves/Piece	Rooting Percentage/%
RM1	0.05	40	89.17 ± 3.82 ^a^
RM2	0.10	40	85.00 ± 2.50 ^ab^
RM3	0.15	40	72.50 ± 4.33 ^bc^
RM4	0.20	40	64.17 ± 8.04 ^c^

Note: Values represented mean ± standard error of three replicates. Different letters within a column indicated significant differences according to Duncan’s multiple range test (*p* < 0.05).

## Data Availability

The original contributions presented in this study are included in the article. Further inquiries can be directed to the corresponding author.

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
