# Peer review of "An Efficient and Streamlined System for In Vitro Regeneration and Genetic Transformation of Paper Mulberry (*Broussonetia papyrifera*)"

_life, 2026, doi:10.3390/life16010078_

Round 1
Reviewer 1 Report (Previous Reviewer 1)
Comments and Suggestions for Authors
The authors have corrected most of the comments. One formal one remains, i.e. unifying the full and abbreviated names of the journal in the References section. This can be done as part of the author's proofreading, and therefore I recommend the manuscript for publication.
Author Response
Comment1: Unifying full and abbreviated names of journals in References:
Response: We have thoroughly checked the References section and unified the format of journal names—using full names for all journals (no abbreviations) and ensuring consistent italicization in accordance with MDPI requirements. For example, “Plant Cell Tissue Organ Cult.” was corrected to “Plant Cell Tissue and Organ Culture”, and “Tree Physiol.” was corrected to “Tree Physiology”.
Reviewer 2 Report (Previous Reviewer 2)
Comments and Suggestions for Authors
Authors improved the manuscript and it can be accepted for publication. Figure 5 "Schematic overview of the hybrid B. papyrifera transformation procedure" can be improved. There is wrong description of the rooting. See: ...root induction mediummedium.
Author Response
Comments 1: Improving Figure 5 (Schematic overview of the hybrid B. papyrifera transformation procedure):
Response: We have corrected "root induction medium" to "rooting medium", which aligns with the terminology used in the manuscript. Meanwhile, we have unified the experimental parameters in the figure legend with the optimized protocols described in the manuscript, ensuring consistency between the schematic and the manuscript content. The revised Figure 5 has addressed the above issues.
This manuscript is a resubmission of an earlier submission. The following is a list of the peer review reports and author responses from that submission.
Round 1
Reviewer 1 Report
Comments and Suggestions for Authors
The authors have made significant changes to the manuscript compared to the first submission and the rejected manuscript.
Nevertheless, I have comments on the manuscript, mainly of a formal nature, because they have an effect on the tone of the manuscript.
line 61 - eucalyptus, poplar and paulownia - Eucalyptus (italics), poplar and Paulownia (italics) would be better;
line 218 - XbaI and KpnI (the part of the enzyme after the microorganism is written in italics by default);
219 - Escherichia coli - italics;
225 - Agrobacterium, A. tumefaciens (italics);
301 - in vitro (italics);
452, 456 - CHS (italics);
559-560 - references are listed without a numerical format;
References section - still not clearly processed, i.e. full and abbreviated journal titles (see Instruction for Authors).
Content comment:
Chapter 2.4.3 requires the addition of the reaction composition (component contraction), temperature and time profile of the reaction or a reference that will contain this information.
Based on the above facts, I recommend the manuscript for publication after major revision and second review.
Reviewer 2 Report
Comments and Suggestions for Authors
The manuscript addresses an interesting and scientifically meaningful topic. Establishing an efficient regeneration and transformation system for Broussonetia papyrifera is valuable both for basic research and for applied work in forestry and biotechnology. The authors clearly attempted to prepare a comprehensive and technically detailed study. However, in its current form, the manuscript is difficult to accept, as it contains a number of fundamental issues that are not compatible with the standards of a scientific article.
Already at the beginning, there is inconsistent use of the full Latin name Broussonetia papyrifera and its abbreviation. The established convention is that the full binomial name should appear once at first mention, followed by the abbreviated form thereafter. This inconsistency occurs throughout the text and should be corrected.
The quality of English is a major concern. The authors often resolve unclear or poorly structured sentences by inserting excessive parenthetical notes, which makes the text harder to read. For example, the sentence “Adventitious shoots with a length of approximately 1.0 cm (induced in the previous step) were inoculated into stem elongation medium” illustrates this problem: the parenthetical phrase is unnecessary and the sentence would be clearer with more direct phrasing. Many similar constructions appear across the manuscript.
Several expressions are simply not acceptable for scientific writing. In the abstract, for instance, the sentence “And then, resistant seedlings of transgenic B. papyrifera were obtained” is grammatically weak and stylistically unsuitable. Numerous other sentences suffer from similar issues. The manuscript requires a thorough revision by a professional English-language scientific editor.
Structural concerns also appear. The section title “3 Results and Analysis” is unusual and not typical for scientific papers; results and analysis are normally integrated within the Results section or clearly separated into Results and Discussion. The current presentation gives the impression of a draft rather than a finalized manuscript.
Because of these cumulative issues, I recommend that the manuscript be returned to the authors for substantial rewriting. A comprehensive linguistic and stylistic revision is necessary before the scientific content can be properly evaluated. Given that the topic is genuinely interesting and relevant, I encourage the authors to thoroughly revise the text with the help of a qualified scientific English editor and resubmit the improved version for further review.
Comments on the Quality of English LanguageMust be improved.